# Designing Internet of Tangible Things for Children with Hearing Impairment

**Sandra Cano** [1,*] **, Victor Peñeñory** [2] **, César A. Collazos** [3] **and Sergio Albiol-Pérez** [4]

1    Escuela de Ingeniería Informática, Pontifica Universidad Católica de Valparaiso, Valparaíso 2340000, Chile
2    Facultad de Ingeniería, Universidad de San Buenaventura de Cali, Cali 760032, Colombia; vmpeneno@usbcali.edu.co
3    Facultad de Ingeniería, Universidad del Cauca, Popayán 190002, Colombia; ccollazo@unicauca.edu.co
4    Instituto de Investigación Sanitaria Aragón (IIS Aragón), Universidad de Zaragoza, 44003 Teruel, Spain; salbiol@unizar.es
*    Correspondence: sandra.cano@gmail.com

**Abstract:** Tangible User Interfaces (TUIs) are a new, non-traditional way to interact with digital information using a physical environment. Therefore, TUIs connect a physical set of objects that can be explored and manipulated. TUI can be interconnected over the Internet, using Internet of Things (IoT) technology to monitor a child's activities in real-time. Internet of Tangible Things (IoTT) is defined as a tangible interaction applied to IoT. This article describes four case studies that apply IoTT to children with cochlear implants and children whose communication is sign language. For each case study, a discussion is presented, discussing how IoTT can help the child development in skills such as: social, emotional, psychomotor, cognitive, and visual. It was found that IoTT works best when it includes the social component in children with hearing impairment, because it helps them to communicate with each other and build social-emotional skills.

**Keywords:** tangible user interfaces; internet of tangibles things; children with hearing impairment

## 1. Introduction

The Internet of Things (IoT) is a technology that allows physical objects to be interconnected through the Internet. Therefore, IoT is a concept related with the transformation digital, where it has integrated several areas, such as: electronics, computers, and multimedia. In 2008, the European Commission [1] define IoT, as: "Things having identities and virtual personalities operating in smart spaces using intelligent interfaces to connect and communicate within social, environmental, and user contexts".

On the other hand, Tangible User Interface (TUI) was initially motivated by ubiquitous computing and augmented reality. In 1995, Fitzmaurice was the first to attribute the term User Interface, he introduces the notion of Graspable Interface, where graspable handles are used to manipulate digital objects [2]. After in 1997, Ishii and Ullmer present a definition of Tangible Bits [3], using real world as a display and as medium to manipulate a digital interface.

An article presented by Angelini et al. [4] associate two terms Tangible User Interfaces and Internet of Things as Internet of Tangible Things (IoTT). Authors present a systematic review of tangible interaction applied to IoT, where they discuss the potential benefits of tangible interaction applied to IoT. Today, IoT is applied to different areas, such as education, health, traffic, agriculture, and public services. For the implementation of physical or embedded objects components such as: sensors, software, and network connectivity are required, which must be incorporated in these objects to store and exchange data.

In several studies [5–9], it is observed that the design of tangible interfaces is incorporating IoT in a context of children. In 2017, Angelini et al. [5] propose a workshop related with IoT and Tangibles Interfaces, where they make a discussion about how best to bridge theoretical, technical, design and human considerations may be taken into account when designing for IoTT (Internet of Tangible Things). Other researchers had created a smart bear [6] with built-in sensors that measures child's heart rate, blood pressure, oxygen saturation, and body temperature and sends the data via wireless technologies to the parent's smartphone.

In 2018, Mahmoudi et al. [7] have developed an interactive learning system for children in color teaching. They incorporated IoT using the Raspberry hardware platform and a color sensor responsible for measuring the frequency of colored light. The information captured from the RGB sensor is sent using the MOTT protocol. In 2019 Ritembruch and Donovan [8] mention that IoT allows the interconnection of physical devices. However, designing a scenario requires further investigate of the interaction between IoT devices and users. The feedback mechanisms and the type of interaction can change according to the type of user.

Other authors have focused on how to teach children to learn IoT. Divitini and Sejer [9], presented a workshop called Make2Learn, oriented to the child learning IoT concepts through the design and development of objects that can be interconnected following the Science, Technology, Electronic, and Mathematics (STEM) concepts. Authors incorporated cards created by Mora et al. [10] as a tool in the ideation stage, composed of 110 design cards, with the aim of supporting exploration and combination of user interface metaphors, digital services and physical objects. The cards are intended to inspire creation or generate new ideas in IoT products focused on the user. Different roles can make use of these cards, such as researchers who can use the cards as brainstorming to know how to include IoT components in their projects. Designers can also integrate the IoT cards into their design methodology, while teachers can use the cards to introduce basic concepts of IoT into the classroom.

Few works are related with tangible interaction principles. TUI involves two terms: user interfaces and interaction. Therefore, the interaction is related with physical world and type of user. In the interaction, the data could be represented through physical objects and manipulated by physically handling the objects. Eva Hornecker and Jacob Buur [11] proposed on tangible interaction using a framework. The framework is focused on the user experience of interaction, so it includes physical and social aspects of interaction. The framework is structured around four themes, as: Tangible Manipulation, Spatial Interaction, Embodied Facilitation and expressive representation. This framework is applied in three case studies, but none is applied to children. Therefore, interaction principles can change for children, more if children have some special needs.

Most children with special needs have low economic resources. Moreover, children with disabilities are among the most stigmatized and excluded groups of children around of world. These children for their disability have less opportunity, in social, education among others. In addition, these children have lower rates of primary school completion that those without disabilities and many cases the technology can help them to develop their learning capacity [12]. However, the growth of technology has led to the emergence of new forms of interaction integrating physical and digital objects interconnecting with others through internet.

An article published by [13], concluded that hearing impairment in Latin America is low priority for national health systems in Latin America, and material and human resources are limited. In Colombia, inclusive schools are limited and the technology continues to be very expensive in the region. In 2009, the National Institute for Deaf People (Colombia) conducted a survey, where it was found that there are 99,693 people with auditory deficit, and more than 50% of the people are in the socioeconomic level low. Therefore, it is not only a problem in Colombia but also in the Latin American region [14].

## 2. Background

### 2.1. Children with Hearing Impairment and Technology

Children have different ways and rates of learning. If they have a disability or disorder, their way of learning may be affected. A child with a disability will acquire cognitive skills at a different rate and using other types of strategies than a child without a disability. Therefore, children with disability or disorder require special education to receive adequate educational development. Hearing impairment is related to impairment of the auditory sense resulting from a partial or total loss of the ability to hear. Therefore, the type of disability is subject to levels of deafness, such as: mild (<40 db), moderate (40–70 db), severe (71–90 db), and profound (> 91 db) [15]. Children with hearing impairment have different communication alternatives: children who do not have access to a hearing aid will communicate using sign language, based on movement and expressions through hands, eyes, face, mouth, and body. So, their first language is sign language, and a second language a written language such as Spanish. If children have a hearing aid as cochlear implants, their main objective is to learn the communication oral for the language acquisition. For children with cochlear implants, the verbalization requires more effort, since they must learn to listen and identify each sound. Therefore, they must learn to use the implant as a means of extracting information. In addition, their learning is different compared with other children without hearing problems.

A learning style can be defined by Keefe [16] as "Cognitive, affective and physiological traits that serve as relatively stable indicators of how students perceive interactions and respond to their learning environments". Cognitive traits are related to the way students structure content, use concepts, interpret information, and solve problems; affective traits are linked to motivations and strongly condition learning levels; and physiological tasks that use physical objects can support a child's cognitive development (supporting the ideas of Piaget) and also allow him/her to take advantage of this real-world experience when interacting with digital information. Another communication alternative is lip-facial reading, where they learn to read lips. Therefore, learning strategies change according the communication channel.

Some work found has proposed recommendations in the design of technologies for children. Even more for designing technologies for children with hearing impairment. An article published by Cano et al. [17] propose a model that allows the identification of a set of principles grouped into three categories: education, game mechanics, and child-profile. These principles are applied in the design of serious games for children with hearing impairment. In 2005, Chiasson and Gutwin [18] proposed an initial catalogue of design principles for children's technology, including three categories as: cognitive, physical, and social/emotional. In the category of physical development, four principles are related with tangible interfaces, as: (1) Children like tangible interfaces because they enjoy being able to physical touch and manipulate the devices, (2) direct manipulatives allow children to explore and actively participate in the discovery process, and (3) physical props and having large input devices encourages collaboration, and (4) surface changes in the design can produce very different physical interactions. Different interfaces emphasize different actions but do not considers whether children have any special needs.

An article published in 2019 by Revelle et al. [19] discusses the use of the traditional interface is often inappropriate from a developmental point of view and can be an obstacle to learning. The use and manipulation of physical objects is a key to the child's learning. Therefore, designing tangible interfaces can be positive for a child with special needs. This can help in the skills development when they interact with physical objects appropriate to the task may be related to the learning environment. According to Piaget [20], where he mentions that manipulation with physical objects can help in the cognitive development of the child. While, Vygotsky emphasized in social interaction for the child development.

The use of technology in children with special needs allows increasing the independence of the child and choose the speed of learning. In addition, in developing technology for special education,

the cost of a given solution and potential it may have in the learning process are considered. Today children are born into world where technology is integrated in the daily life. Studies show a positive trend in the relationship between learning and technology integration [21]. Ozgur and Seyhan [22] believe that the use of technology can impede in children's social, emotional, physical, and cognitive development. However, if activities are monitored and integrated in the classroom, and children can be supported by the teacher, this can be positive.

### 2.2. Child-Computer Interaction and Tangible User Interfaces

Human-Computer Interaction (HCI) is an area that focuses on the interaction between a person and machine. Child-Computer Interaction (CCI) is a sub-field in HCI that relates concepts between children, computational, and communication technologies. Therefore, it involves input and perspective with multiple scientific disciplines to design an interactive system for children. In 2011 Read and Bekker [23] define CCI as "study of the activities, behaviours, concerns and abilities of children as they interact with computer technologies, often with the intervention of others (mainly adults) in situations that they partially control and regulate".

In 1980 Papert started with the computer technology for children. He began to investigate how children could benefit from the technology as an assistive tool [24], with the design of a product called Logo [25]. Papert developed an approach, following Piaget's constructivism, that consists of placing challenges to children in such a way that these can be solved by developing programs with Logo. The Logo program was one of the first interfaces in which the concept of interaction changed: it was no longer a simple interaction with traditional computer devices, but designed another type of non-conventional interaction to communicate with the computer. Therefore, Logo began the creation of technological tools that support children's learning, developing exploration, and interaction skills.

There are TUIs developed with a variety of physical objects. However, they are limited for a type of user with special needs, such as children with hearing impairment. Druin et al. [26] comment that children want in technology: control, social experience, and expressive tool. Therefore, technology must produce curiosity, motivation for repetition, and need for control.

Nowadays, some problems include CCI and Tangible Interfaces. It is related to how to evaluate Tangible Interfaces for children. There are two ways a child can interact with the technology: physically and digital. It is now an area where many studies have been proposed with new methods to evaluate technologies with children. However, most of these methods created are for children without special needs. Read and Bekker [23] take into account that the CCI must consider the physical sizes and abilities, memory and processing abilities, and the ability of children to read (e.g., deaf children), but has the additional task of understanding the changes and diversity in this space.

### 2.3. Tangible User Interfaces and Internet of Things

Advances in technology are creating new opportunities, services, and mechanisms to provide a better quality of life. The way all users can communicate over long distances is through the Internet. Therefore, connectivity between different elements is of great interest for the development of prototypes oriented at storing information in the cloud or connecting physical objects.

TUI was initially motivated by ubiquitous computing and augmented reality, as a more natural way to manage a device. It is believed that physical action is important for learning, and tangible objects are a way for the user to learn while interacting with physical objects. Ulmer and Ishii, from the Tangible Media Group at MIT Media Lab, define TUIs as devices that give physical information as representations and controls of the computational data [25]. Historically, children have always interacted with physical objects in the classroom to learn some functionality.

Ulmer and Ishii [27] describe some aspects that should be taken into account when designing a tangible interface: (1) linking physical representations and their digital information, (2) designing interactive control modes, considering tangible representations, and (3) perceptual coupling of tangible representations to intangible dynamic representations. Thus, it can be said that tangible interfaces allow

physical representations through technology to make more real the scenarios that can be interconnected using IoT technology.

The potential of IoT for children is growing. One example is Teddy the Guardian [6], a smart teddy bear with an internal accelerometer waiting mode when it is quiet. Teddy also has a temperature sensor that measures the temperature of the child. On making a movement with the bear, it wakes up, remaining in an environment and the temperature of the child. The teddy bear is connected to a mobile application where it sends data, and professional personnel can monitor physiological responses of the child.

Other work is proposed in 2018 by Cano et al. [28], where they used a teddy bear for teaching STEM education, where a nano-Arduino, RFID card reader and Neopixel have been added to the bear as feedback response, according to whether actions performed are correct or not. The bear connects to a mobile application, which includes three areas: Literacy, mathematics, and programming.

It is important to mention that the type of Tangible Interfaces as Table-Top can be high cost. However, our goal is to design products using low-cost electronics and reusing existent toys.

## 3. Case Studies

The following case studies are tangible interfaces where IoT technologies have been included and used in children with hearing impairment in different contexts, such as: education, social, and therapy. The participants were children with hearing impairment who benefit from hearing aids, such as cochlear implants, or who lack hearing aids and communicate through sign language. The children study at Institute for Blind and Deaf Children in Valle del Cauca, Colombia. There are also children whose communication channel is sign language, from the Special Sense Therapy Institute of Club Leones (ITES), in Cali, Colombia, were aged three to twelve. For each of these evaluations carried out with the children, informed consent was obtained from parents. The ethical principles established by the Helsinki declaration were also followed [29].

For each case study was applied some techniques to know the child-profile, as: ethnography, direct observation, interviews, and survey.

### 3.1. Case Study 1: Cognitive Rehabilitation

Based on the analysis carried out, a set of mini-games aimed at stimulating cognitive processes is proposed [30]. Therefore, a tool is proposed that helps stimulate a number of cognitive processes in children with hearing disabilities, called Stimulating with PhonaTIC (Figure 1), with the aim of capturing information about each process, such as: visual memorial, selective attention, auditory perception, perceptual discrimination, and spatial orientation.

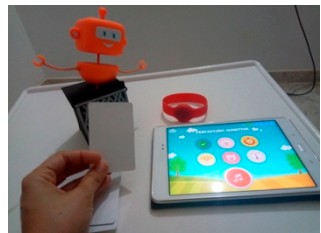 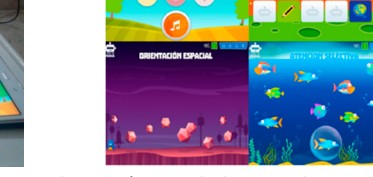

(**a**) Physical Object with RFID cards　　(**b**) Mobile Application

**Figure 1.** Tangible interface Stimulating with PhonaTIC for children with a cochlear implant.

The interface employs a range of activities such as visual memory, selective attention, auditory perception, and spatial orientation. For the physical and digital interaction part, we worked on visual memory and auditory perception activities. In the case of visual memory, the user must associate objects with the word, so the digital application will show the word must associate it in the RFID cards

with the figure that represents that word. Auditory perception is related to the child hearing sounds from different animals and selecting the animal to which the sound corresponds.

The information collected relates mainly to the measurement of their learning. Therefore, it was decided to capture several indicators such as the time it takes to carry out each activity and the number of successful or failed attempts. The application is made in Android Studio, and web services in PHP are used to send the data to the database. Therefore, the mobile application consumes the web services when the information is captured while the user interacts with the application.

### 3.2. Case Study 2: Interactive Toy

A teddy bear called Tobi (Figure 2) is proposed which is conditioned with sensors and other electronic components, to become a low-cost interactive toy for children with special needs, supporting Science, Technology, Engineering, and Mathematics (STEM) education as a fundamental pillar of learning.

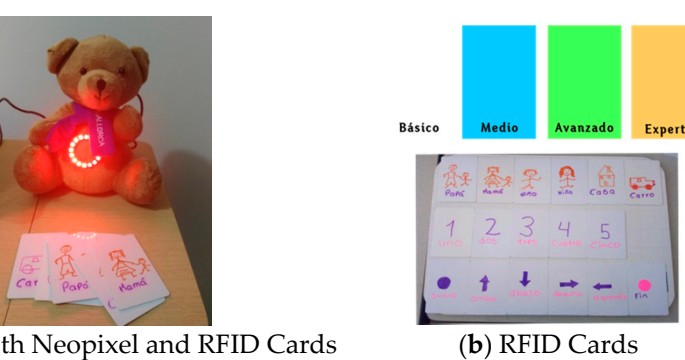

(**a**) Bear with Neopixel and RFID Cards      (**b**) RFID Cards

**Figure 2.** Interactive toy called Tobi using Science, Technology, Engineering, and Mathematics (STEM) methodology. In [28].

Based on the concept of tangible interfaces, the starting point here is an appreciation of the toys children have and that technology can be integrated into them to become interactive. A hardware structure was designed to be included inside the body of a teddy bear, where the low-cost MFRC522 RFID sensor, Gyroscope GY-50, and HC06 Bluetooth sensor were added to Tobi's body to establish communication between the bear and a mobile application. The tangible interface has two ways of visual representation supported by the teddy bear and the mobile application. The action that the child must perform is to bring the card close to Tobi's chest and the application will validate if it is correct. The effect is that if it is a correct answer, the LEDs located in the bear's chest turn red, as if it will symbolically represent the bear's heart and that it is happy. If it is not correct, the LEDs have an effect of turning on and off, representing symbolically as if it were annoyed [28]. The IoT technology was integrated, saving the data in a database by connecting through web services in PHP, with the aim monitoring the different activities of child, as: number of errors, time, and levels completed.

This toy was created with the purpose of involving the STEM methodology for children without special needs, where activities related to literacy, mathematics, and computational thinking are integrated. The IoT technology was integrated, saving the data in a database by connecting through web services in PHP.

### 3.3. Case Study 3: Electronic Glove

In 2017, a group of researchers presented at ITES the design of an electronic glove for teaching vowels through the language of fingerprint for deaf children [31]. They designed a tangible interface, which includes a physical and a digital part. The physical part is a non-traditional input device, a glove with sensors to recognize the gesture of each vowel. The glove is connected via Bluetooth to a mobile application, where the child must perform different finger actions to represent a vowel (see

Figure 3). It is important to mention that fingerprint language is used for spelling in written work. The IoT technology was integrated, saving the data in a database by connecting through web services in PHP, with the aim that teachers can monitor the different activities of child, as: number of errors, time, and tasks completed.

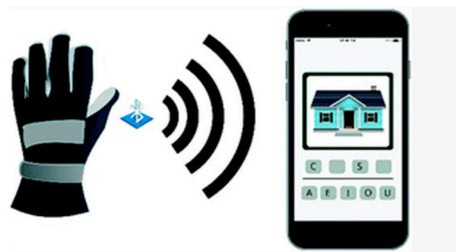

**Figure 3.** Design of an electronic glove for teaching vowels for deaf children. In [31].

The evaluation was carried out with five deaf children aged six and ten, whose communication channel is sign language. To evaluate the prototype, the metrics of number of errors and time taken for completed tasks were considered. Their evaluation was more quantitative, the child experience when interacting with the glove was not evaluated, but was considered variables related to cognitive effort. Meanwhile, in the mobile application tasks, a set of words had been selected, for which the children had to complement the vowels to form the correct word.

### 3.4. Case Study 4: Problem-Solving

The design of the tangible interface called Perdi-Dogs is aimed at the acquisition of computational thinking skills for children with cochlear implants [32]. A physical interface is proposed, which is a ladder board game (Figure 4), whose objective is for the child to find the best solution to the problem at hand. The mission that is recreated aims for each dog to reach his home, so that the child must make correct decisions to overcome the obstacles that arise. The digital interface is roulette type, which has the function of giving each child a turn in which the number of squares that the dog must advance is obtained randomly. The way to connect the physical and digital interface is though physical cards with a QR code. This code is read by mobile application. Children can interact in group, and they support each other and are not inhibited in carrying out the activity. they are also not afraid of making mistakes. The data information is saved in a database, which the mobile application is connected to a web service.

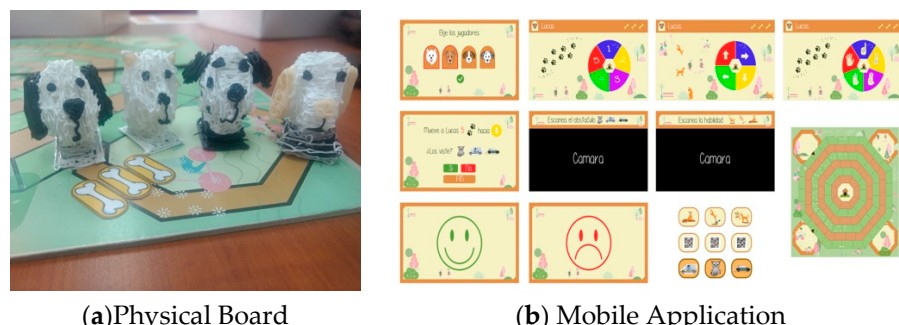

(**a**)Physical Board　　　　　　　　　　(**b**) Mobile Application

**Figure 4.** Tangible interface Perdi-Dogs.

It is important to mention that in each of the case studies presented, each child must access a user profile, where data is collected according to their profile. Today, it is in discussion to integrate the IoT into the digital application using a cloud service, such as Firebase database in the Google Cloud. The data can access via a Google service, where the data is stored in JSON.

## 4. Discussion

Keller [33] presents a technique to model the workload, which includes rating scales developed by McCrasken and Aldrich [34]. The scales provide an assessment of the degree of utilization of each resource. Each scale value includes a description and value according to the use of the resource component. The higher the scale value, the greater the degree of use of the resource component. The components measured according to the workload, are: visual, auditory, cognitive and psychomotor. Therefore, this scale called Visual, Auditory, Cognitive, and Psychomotor (VACP) (See Table 1) is applied to each case study (See Table 2), but the auditory component is not consider, since the participants are children with hearing impairment, and the interest is to observe how many workloads other components can associate for each proposed case study. The components allow a simulation to provide quantitative predictions of workload during interaction with the system.

**Table 1.** Model Visual, Auditory, Cognitive, and Psychomotor (VACP) used by Keller [33] based on the dimensions of Wickens.

| Scale Value | Scale Description |
|---|---|
| **Visual** | |
| 0.0 | No visual Activity |
| 1.0 | Visually Register / detect (detect occurrence of image) |
| 3.7 | Visually discriminate (detect visual differences) |
| 4.0 | Visually Inspect/Check (discrete inspection/static condition) |
| 5.0 | Visually Locale/Align (selective orientation) |
| 5.4 | Visually Track/Follow (maintain orientation) |
| 5.9 | Visually Read (Symbol) |
| 7.0 | Visually scan /search/monitor(continuous/serial inspection, multiple conditions) |
| **Cognitive** | |
| 0 | No Cognitive Activity |
| 1.0 | Automatic (simple association) |
| 1.2 | Alternative Selection |
| 3.7 | Sign/Signal Recognition |
| 4.6 | Evaluation/Judgment (consider single aspect) |
| 5.3 | Encoding/Decoding, Recall |
| 6.8 | Evaluation/Judgment (consider several aspects) |
| 7.0 | Estimation, Calculation, Conversion |
| **Psychomotor** | |
| 0.0 | No Psychomotor Activity |
| 1.0 | Speech |
| 2.2 | Discrete Actuation(button, toggle, trigger) |
| 2.6 | Continuous Adjustive (flight control, sensor control) |
| 4.6 | Manipulative |
| 5.8 | Discrete Adjustive (rotary, vertical thumbwheel, level position) |
| 6.5 | Symbolic Production (writing) |
| 7.0 | Serial Discrete Manipulation (keyboard entries) |

**Table 2.** Applying VACP for each case study.

| Case Studies | Visual | Cognitive | Psychomotor | Total |
|---|---|---|---|---|
| Cognitive Rehabilitation | 5.9+1.0 | 3.7+1.0 | 2.2 | 12.8 |
| Interactive Toy | 5.9 | 1.0 | 4.6 | 11.5 |
| Electronic Glove | 5.4 | 3.7 | 2.6 | 18.9 |
| Problem-Solving | 4.0 | 4.6 | 1.0+2.2 | 11.8 |

The purpose of the model is to determinate the level of workload accumulated during the use of each IoTT. When the child interacts with the IoTT, it is necessary to understand the effort it may

involve for the child. However, child factors such as age and technology-experience have not been taken into account.

The case studies concerned the design of tangible interfaces applied to different contexts of use for children with hearing impairment. The integration of the IoT can help provide access to information to people who want to monitor the different activities of the child, as well as being able to see the progress of the child in real-time. On the other hand, the use of IoT objects can help to explore different ways of interacting and also how to integrate feedback modalities, such as haptic vibrations. This modality for children with hearing impairment is positive, because children may have a low level in literacy, so they cannot read text messages. Another way is by using movement gestures, such as the electronic glove. Children can easily process information using two modes of feedback through the senses, as a visual and tactile response. In addition, when interacting with a TUI they do not feel evaluated. Therefore, the teacher can observe different captured indicators that reflect progress toward goals. Some indicators may help to monitor the advance in the activity and perhaps adjust your learning style or speed. Therefore, the data collected allows for the study of the child's behavior patterns.

When children interact with real objects with other children their motivation grows because they do not feel evaluated. It was observed that when these types of tangible interfaces are evaluated in groups, rather than individually, children provide more information because they are less inhibited in their answers. It was also identified that when interfaces have a learning objective for the child, the interaction experience should be taken account, since the main goal is to carry out a task oriented in an educational learning context. If the experience is not positive, the learning will be longer, even causing a cognitive effort.

Sensory processing can influence the social and emotional development for Children with hearing impairment. Children are deprived of auditory information, and their interactions are reduced. These children process information differently than a hearing child. Therefore, they need to receive appropriate interactive feedback. In the interactive bear (case study 2), there were two types of visual and auditory feedback. The auditory feedback was that through the mobile application. The instructions were carried out in an auditory way where the child had the task of associating a sound with a pictogram. The sound produced by the application used Google's text-to-speech service, and it did not work correctly, the children did not hear the pronunciation very well and the teacher always had to approach the child to repeat the word. Therefore, it was necessary to record the sounds and eliminate the auditory feedback.

Table 3 shows a comparison of the four case studies by type of skills (emotional, social, psychomotor, visual, and cognitive) that might help in the child's development. It should be noted that the assessment carried out is subjective and qualitative, according to the observations obtained while the children were interacting with the interfaces. Therefore, Table 3 shows that the cases applied arouse feelings (curiosity) in the children, but do not work the emotional development. All cases show emotional feedback associated with two emotional states (joy and sadness), since the teachers evaluate the children with only two emotional states. When the child interacts with an IoTT, he/she must collect information from the physical environment, explore and manipulate objects. The electronic glove (case study 3) used for deaf children, whose communication channel is sign language. This device is used to spell words. In deaf children, it is used as a dactylology, which helps in the phonological code, and can be an alternative for independent memorization, that is it recognizes if it is correct or not.

**Table 3.** Comparison to each case study in the skills development for children with hearing impairment.

| Skills/Case Studies | Feelings/Emotions | Psychomotor | Visual | Cognitive | Social |
|---|---|---|---|---|---|
| Cognitive Rehabilitation | X | X | X | X | |
| Interactive Toy | X | X | X | X | |
| Electronic Glove | X | X | X | X | |
| Problem-Solving | X | X | X | X | X |

Cognitive development has been studied in various ways, especially in hearing impaired children, because their information processing is different from that of a hearing child. Cognitive development is how a child perceives, thinks, and understands the world. Therefore, sensation, perception, memory, and learning are topics of interest for research. An article published by [35] developed a model of cognitive interaction for analytic evaluation of inclusive design. The authors state that experience is a critical factor in how easy it is to learn a system. Therefore, when an interface has different responses associated with it at the same time, they may have a negative experience, as occurred with case study 2. Another factor that was observed in the feedback applied to the interactive toy. When the child did not perform the activity correctly, the LEDs located in the bear's chest turn green, having an effect of turning on and off, representing symbolically as if it were annoyed. However, the children did not understand the symbolic representation of the incorrect response. Therefore, in the second interaction with them, the negative response was omitted leaving only the positive response. They understood that if the LEDs did not light up it was because something was not right. Case study 2 worked with children with cochlear implants aged 3–5 years old in auditory therapy. These children are beginning the process of language acquisition.

The Cognitive Rehabilitation Interface (case study 1) has an interaction between the character and a mobile device using RFID cards. The mobile application has a number of activities such as: memory, selective attention, auditory perception, and spatial orientation. Children with cochlear implants must learn to listen, so they must learn to recognize sounds, and they also have difficulty with selective attention [36]. While, deaf children who communicate with sign language must develop spatial attention skills to a greater extent than those with normal hearing. However, the children would expect physical character to make some interaction by performing the activity correctly. When they noticed that the interaction was more on the side of the mobile application they were not very excited, they asked why the robot was not doing anything or not talking.

All the previous case studies work on three skills (feelings, physical, and cognitive), but the social one is only worked on in case 4 (problem-solving), as it allows the child to integrate with other children to perform the activity. The social component is very important in the child because when he/she performs the activity he/she no longer feels inhibited in his/her response and does not wait for the approval teacher. At the beginning of the physical game a dialogue was established with the children explaining the story and their mission. This part is important to contextualize the child and awaken oral comprehension with them. Then, 3 sessions are held with the children to evaluate the experience each time they play.

In the first experience with the interface it was observed that the children with cochlear implants were confused, they did not know in which direction to move the character, nor what to do when there was an obstacle. The teacher intervened to explain what they should do, but they still needed the teacher's approval to know if they were doing it right. Some found it more difficult to understand how it worked, but they saw the other children and trusted them more than the teacher. Similarly, this first experience for some children required more mental effort.

In the second experience with the interface it was noted that the children were already working independently. They supported each other when they had to interact with the mobile application. Therefore, if the other child did not know which direction to take, the other child would point him in the right direction. When they faced an obstacle, they were afraid, because they had to know what decision to make, otherwise the number of their character lives would decrease. However, in this second experience they still found it difficult to orient themselves to the left and right. Finally, in the third experience they were more independent, more confident in their decisions.

In Vygostky [37], social interaction is an important way for children to learn available knowledge. Parents, adults, teachers, and peers all play important roles in the process of ownership of children's learning. The children in case study 4 used peer conversations, exchanged ideas and received information from the teacher. According to Zigler et al. [38], "The promise of play-and its many other well-documented benefits-extends significantly beyond the development of literacy, arithmetic, and

science skills. Play contributes to the emotional, intellectual, physical, social, and spiritual development of the child in ways that cannot be taught through instruction. They must be experienced and play is the natural, built-in way children accumulate that *experience*".

IoT in schools is integrating personalized learning. Therefore, IoT can help the teacher to have results for each child at the end of the class, such as: number of mistakes, completed tasks, duration of time, description of the mistakes. Therefore, they can monitor the activities of each child, and they will be able to make an analysis of the child's progress and to measure the progress of the leaning child in real-time.

## 5. Conclusions

The growth of technology is creating new opportunities to provide individuals with a better quality of life. Advances are having an impact on the different ways in which there can be an interface between a person and technology. The internet has made it possible for people to interconnect and use of IoT has meant that devices or objects can do so as well. Tangible interfaces and IoT technologies can support different types of contexts and populations, even more so if children with special needs are involved.

It was observed that when these types of tangible interfaces are evaluated in groups, rather than individually, children provide more information as they are less inhibited in their responses. The questions that are asked to the children should be formulated very clearly. Therefore, instruments of the type on which the child should give a rating were not considered, as many of them proved to be unable to handle a rating scale from 1 to 5. It was also identified that they did not understand many questions because they were in the process of acquiring language. They also did not recognize all facial expressions related to emotions. The questions asked included Did you find it difficult? Did you have fun? Did you understand the task you are supposed to do?, but they were all positive answers. For this reason, they were not included in the article, as this was very subjective information.

It was also noted that what motivated the children most was finding an unusual way for this type of interface to interfere with the Tablet than what they were physically doing had digital feedback. Therefore, there is a lack of studies describing what elements are important to children experience of this type of tangible interface, especially when working with children who have physical limitations. The intention in future work is to evaluate in a more quantitative way the interaction with the tangible interface as well as the quality of the system. In this type of interface there are two types of users, the teacher who uses IoT technology and the child who interacts with the physical-digital interface.

Therefore, IoT can help provide access to information and interaction with others, such as children with/without special needs. It can continue to support the creation of an inclusive environment with better access to information and interaction with others, and like these kinds of tangible interfaces, it can support children operating within the focus of a real—and at the same time digital—environment.

The design of this type of IoTT works best when working on the social aspect. However, in this type of interface where the physical and digital are involved, it is difficult to evaluate the experience. Therefore, it is necessary to study and analyze techniques that can help measure the child experience in an environment where various artifacts are integrated.

**Author Contributions:** Conceptualization, S.C. and V.P.; Methodology, S.A.-P.; Software, S.C.; Validation, S.C., V.P. and C.A.C.; Formal analysis, S.C.; Investigation, S.C.; Resources, S.C.; Data curation, S.C.; Writing—original draft preparation, S.C.; Writing—review and editing, S.C.; Visualization, S.C.; Supervision, S.C.; Project administration, S.C. All authors have read and agreed to the published version of the manuscript.

**Funding:** This research received no external funding.

**Acknowledgments:** Institute of Blind and Deaf Children and the Special Sense Therapy Institute of Club Leones (ITES), in Cali, Colombia.

**Conflicts of Interest:** The authors declare no conflict of interest.

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
