# Peer review of "Designing Internet of Tangible Things for Children with Hearing Impairment"

_information, doi:10.3390/info11020070_

Round 1

Reviewer 1 Report

The paper addresses a very important topic - aiding children with hearing disabilities. The authors do a good job highlighting the importance of the issue and providing some review in the field.

However, I fail to see how the current paper contributes anything additional to the previous work done by the authors and their referenced publications. It is perfectly fine to have a journal publication that extends conference papers/reports, but the publication must also have its own significance. The authors, however, just tell in short about 4 case-studies (mentioning, but not showing any data), but do not provide any comparisons or aggregating conclusions. The readers can just read the previous publications by the authors and get much more information from them.

I would highly recommend to reconsider the approach and focus the current paper on something that unites all the previous work / case studies. E.g. what approach/tool is more effective, or possibly in which context what works better?

The authors should also carefully check and correct the English of the paper - currently it has grammatical errors that hinder understanding. E.g. in p. 3: "But the principles proposed are oriented a technology type and children group with special need specific."
Sometimes words seem to be missing, e.g. p. 3: "In Chile published a study by [14]..."
Formatting of the text should be made consistent (e.g. in p. 2, the main text style suddenly changes).

I hope that the authors will re-submit their paper, and I'm ready to review the improved version.

Author Response

The current paper corrections were made to both the English (grammatical errors) and term "Internet of Tangible Things". 

There is a new section called results taking VACP scale of Keller applied for each case study. In discussion there is a table, where it was compared the four case studies taking into account the skills development, as: Pyschomotor, visual, cognitive, emotional and social. According,  the skills development is stablished a discussion for each case study, where qualitative results obtained are discussed.

Finally,  it mentions which approach is better and which interface.

Reviewer 2 Report

The manuscript is a presentation of four use cases describing how Internet of Tangible Things design methods have been applied to systems for children with hearing disability. The topic is really interesting and the applications of IoTT can improve healthcare, education and learning settings. 

While the topic is really interesting, the contribution of the paper is unclear to me. I would expect such a paper to either be a survey on recent works for IoTT and children or a proposed framework for designing such systems. In its current version, I find this paper incomplete.

Some comments you could consider:

The terminology is Internet of Tangible Things - not Internet of Things Tangible. It should be edited through the whole manuscript. Moreover, edits of English language is required.  The discussion and conclusion sections should be extended to include both "lessons learned" from the use cases, as well as future directions, open challenges, research trends in the context of IoTT and children with hearing disabilities.

The manuscript includes useful information in the specific area but, in my opinion, the material and the structure/presentation could be improved to make the contribution of the paper clearer and stronger.

Author Response

(The authors gave the same response as above.)

Round 2

Reviewer 1 Report

The added material generally satisfies the comments I made in my previous review, so I recommend accepting the paper.

However, the authors are strongly advised to further improve the English writing in the paper. Particularly, the new Results chapter is problematic. E.g. in p. 8: "According scale values proposed by Keller, in Table 2 shows VACP for each case study.", "the auditory scale is not consider", etc.

Author Response

The article was reviewed in the required English language and style. In addition, the abstract was written in accordance with the comments received by the reviewers.

Reviewer 2 Report

The second version of the manuscript is significantly improved. The author took into consideration the comments and added a new results and discussion section.

Some minor comments:

the abstract needs editing -- a better definition of TUI and one sentence about the new sections proofreading and English edits are still required the result and discussion sections could be merged into a single section, e.g., Discussion or Remarks, the "results" title is misleading

Author Response

(The authors gave the same response as above.)
